# Can Photoselective Nets’ Influence Pollen Traits? A Case Study in ‘Matua’ and ‘Tomuri’ Kiwifruit Cultivars

**DOI:** 10.3390/plants13121691

**Published:** 2024-06-19

**Authors:** Helena Ribeiro, Nuno Mariz-Ponte, Sónia Pereira, Alexandra Guedes, Ilda Abreu, Luísa Moura, Conceição Santos

**Affiliations:** 1Department of Geosciences, Environment and Spatial Plannings, Faculty of Sciences, University of Porto, 4169-007 Porto, Portugalaguedes@fc.up.pt (A.G.); 2Earth Sciences Institute (ICT), Pole of the Faculty of Sciences, University of Porto, 4169-007 Porto, Portugal; 3Department of Biology, Faculty of Sciences, University of Porto, 4169-007 Porto, Portugal; 4Associated Laboratory for Green Chemistry (LAQV-REQUIMTE), University of Porto, 4169-007 Porto, Portugal; 5Centre for Research and Development in Agrifood Systems and Sustainability (CISAS), Polytechnic Institute of Viana do Castelo, 4900-347 Viana do Castelo, Portugal

**Keywords:** germination, morphology, protein content, Raman spectroscopy, soluble sugars, viability

## Abstract

The increasing use of photoselective nets (PNs) raises the question of their influence on pollen traits. We aimed to evaluate the effect of PNs (yellow, pearl, and grey) on the pollen of ‘Matua’ and ‘Tomuri’ *Actinidia deliciosa* cultivars. The pollen size and the exine were studied with a light microscopy and a scanning electron microscopy, and the fertility was analysed by a viability assay and in vitro germination. The total soluble proteins (TSPs) and sugars (TSSs) were quantified by colorimetric assays. The molecular structure of the pollen grain’s wall was analysed by a Raman spectroscopy. The pollen from the plants under the PNs had a larger width and area and a lower germination rate. No significant changes were observed in the exine’s microperforations. The TSP and TSS contents were influenced by the cultivar and PNs (particularly the pearl PN). The Raman spectra of the pollen from the plants grown under the nets presented some bands that significantly shifted from their original position, indicating differences in the vibration modes of the molecules, but no overall changes at their structural or organisation level were found. Our study showed that the PNs could influence several pollen traits, with the pearl PN inducing greater modifications. Our results also support the idea that cultivars affect the outcome of some characteristics.

## 1. Introduction

Kiwifruit has been assuming an increasingly substantial position in the world’s agricultural panorama, considering the sustained growth of its production year after year in the last two decades. Indeed, global kiwifruit production has more than quadrupled from 1999 to 2019, now exceeding 4 million tonnes annually [1].

As seen in several fruit tree crops, covering the orchards with protective nets is raising interest among kiwifruit farmers [2,3]. It mainly results from farmers’ necessity to protect their crops from adverse environmental conditions, because those nets can offer protection against hailstorms, excessive solar radiation, or pests [3]. However, amongst those nets, photoselective nets (PNs) have been generating special interest due to their ability to combine protective functions with the promotion of desired light-regulated physiological responses [3,4]. Such responses result from the altered light quantity and quality that depend on the compositional characteristics of each PN [3,5]. Along with the increasing use of PNs worldwide in several fruit tree crops over the last few years, research in this sphere has been proliferating. Kiwifruit is no exception [6]. However, similar to what has been observed in other fruit tree crops, studies have focused mainly on the effects of PNs on vegetative growth [2] and fruit quality [7,8].

As in the case of many fruit tree crops, satisfactory production is highly reliant on pollen quality, which, in turn, can be influenced by environmental and phytotechnical factors. In fact, proper fructification can only be achieved if an effective transference of pollen from the anthers to the pistils occurs and if the pollen grains that reach the stigmas are viable, exhibit germinative capacity, and originate pollen tubes able to fertilise the ovules [9]. In particular, marketable kiwifruit is strictly dependent on efficient pollination [10], and pollen fertility is crucial to obtaining fruits with the required size and weight for market demand [11].

Although urging more elucidation, Leite et al. [12] have already reported negative consequences of an anti-hail net shield on the pollination in apple trees. Furthermore, Blakey et al. [13] and Stones et al. [14] have demonstrated a reduction in bee activity under shading nets in avocado orchards. However, to our knowledge, the influence of covering nets, such as PNs, on aspects of pollen, such as its morphology, fertility, and molecular composition, still needs to be characterised. Nonetheless, the development and performance of pollen grains is sensitive to local environmental conditions. The consequences of abiotic stressors, such as heat, drought, or nutrient availability, on pollen traits have been investigated, particularly with crops’ resistance/adaptation to climate change, plant conservation, and breeding programmes in mind [15]. For example, it has been reported for many crops that reduced pollen fertility and increased pollen size are the effect of heat, as well as several forms of proteome and metabolome re-programming [16]. Temperature and humidity are among the major factors that can alter some biochemical characteristics of the pollen grain, such as the irreversible loss of the cell membrane permeability or the inactivation of protein enzymes or growth hormones [15,17].

Therefore, we hypothesize that pollen traits could also respond under the influence of PNs, due to the manner that this agricultural technology influences the level of shading, the light scattering enrichment, and the proportion of spectrally modified and unmodified radiations reaching plants and alters the meteorological condition under them [18].

This research aimed to study the influence of PNs on pollen’s morphology, fertility, and biochemical properties and to give the first scientific information on the effects of PNs on the kiwifruit pollen.

## 2. Results

The *Actinidia deliciosa* pollen presented an average size (length (P) from 32 to 33 µm; width (E) around 16 µm; and total area occupied by the pollen between 410 and 435 µm^2^) and prolate shape (P/E ratio around 2 µm), with four to six microperfurations of the exine per µm^2^ (Appendix A). The morphometric parameters indicated that the variable cultivar did not present a statistically significant influence on the average values of the measured parameters. On the contrary, the pollen collected from the plants under the PNs, particularly the pearl and grey, had a significantly bigger width and total area occupied by the pollen than those collected from open field conditions (Appendix A). A significant interaction between the cultivars and the PNs was observed for the P/E ratio (Figure 1).

The average pollen viability of the Matua cultivar ranged between 71.5 ± 4.8% for the plants grown under the pearl net and 83.1 ± 4.0% for the plants under the yellow net. The plants in open field conditions presented an average pollen viability of 77.9 ± 3.6%. For the Tomuri cultivar, the average viability values ranged from 70.0 ± 2.4% for the plants grown under the pearl net to 77.0 ± 3.6% for the plants under the grey net. The average pollen viability in open field conditions was 76.2 ± 4.5% (Appendix A). Concerning the pollen germination rate, it was lower than the viability. The Matua cultivar presented an average pollen germination ranging from 69.7 ± 3.2% in the plants grown under the pearl net to 77.8 ± 3.1% for the plants in open field conditions. For the Tomuri cultivar, the average germination values ranged from 66.5 ± 4.7% in the plants grown under the pearl net to 74.1 ± 2.0% for the plants in open field conditions.

The cultivar and the PNs significantly influenced the average pollen viability and germination, but no significant interaction was seen between these two co-factors (Figure 1). For the different cultivars, the Matua presented a significantly higher average pollen viability and germination rate than the pollen viability and germination rate of the Tomuri cultivar. Regarding the PNs, only the pearl net had a negative influence on the pollen viability, while a significant reduction in the average pollen germination rate induced by the yellow and pearl PNs was observed, with the highest decrease registered in the plants grown under the pearl net (Appendix A).

The total soluble protein (TSP) and sugar (TSS) contents presented the most variable results with different trends between the cultivars (Figure 1, Appendix A). For the Matua, a decrease in the TSP was observed in the pollen from the plants under the nets, varying between 145.4 ± 11.9 µg/mL in open field conditions and 130.2 ± 14.0 µg/mL under the yellow net. For the Tomuri pollen, either an increase (yellow 149.3 ± 6.6 µg/mL and grey 147.0 ± 9.5 µg/mL) or a decrease (pearl 132.2 ± 6.7 µg/mL) was observed. The two-way ANOVA analysis showed no significant influence of the cultivar or the nets in the average TSP content variations (Figure 1). Nonetheless, a significant interaction between the two variables was observed, meaning the factor of the cultivar impacts the net influence on the TSP content.

Concerning the TSS, a decrease in the content of the Matua pollen collected from the plants grown under the yellow net (121.3 ± 7.4%) and grey net (134.1 ± 8.5%) was observed, compared with the pollen from the open field conditions, which was similar to the one in pollen from the pearl net (142.4 ± 8.2%). The opposite behaviour was observed for the Tomuri, with an increased TSS content in the pollen from the plants grown under the nets, varying between 122.9 ± 8.5% in the plants grown in open field conditions to 143.7 ± 10.5% in plants grown under the pearl net. The statistical analysis showed a significant influence of the PNs in TSS variations, as well as an interaction between the cultivars and the PNs.

The Raman spectra of the pollen from the Matua and Tomuri cultivars developed under open field conditions were similar (Appendix A). However, the pollen spectra from the plants grown under the nets were significantly shifted overall from their original position with the bands’ wavenumbers of ≈1006–1010 cm^−1^, of ≈1036–1039 cm^−1^, of ≈1127–1133 cm^−1^, of ≈1174–1177 cm^−1^, of ≈1207–1211 cm^−1^, and of ≈1609–1612 cm^−1^ (Figure 2, Appendix A).

Similar to the previously reported details for the TSP and TSS, a cultivar-dependent effect was observed in the registered modifications. The Matua samples showed a significant shift towards a smaller wavenumber in all nets, with the pearl net inducing the greatest change. For the Tomuri, the shift occurred towards a higher wavenumber, but, in the spectra of pollen collected under the grey net, the bands’ average position, although higher, was not significantly different from the control. In this cultivar, the yellow and pearl nets induced the greatest change.

## 3. Discussion

The pollen morphometric analysis indicated a significant increase in the width and area when produced under the distinct photoselective nets (PNs), with similar behaviour in both cultivars. This size change could be related to the pollen grain dehydration status during the last phases of maturation in the anther or at anthesis, which may vary according to the environmental conditions the plant is exposed to. The possibility of pollen entering into complete or partial developmental arrest is documented as a defence mechanism for extending its viability during the pollination time. This arrest is thought to be associated with the acquisition of desiccation tolerance (DT), with water loss, to more or less of an extent, depending on the environmental conditions at the dispersal time [19]. Under the nets, it is expected that the existence of higher relative humidity conditions [20] will cause the pollen produced to lose less water than the pollen formed in open field conditions. The larger size of the pollen, associated with less wind under the nets, might lead to a faster pollen deposition and, therefore, impact the pollination process. However, more hydrated pollen can be metabolically active, germinating quickly upon landing on a stigma [19].

Our results point out a fertility reduction under the nets, being significant in the pearl (viability and germination) and yellow (germination) PNs, with the cultivar having no significant influence on the outcome of the different PNs. An association has been reported between reduced male fertility and a reduction in the incident radiation in the months preceding the anthesis in wheat but no effect from air temperature [21]. The use of PNs allow the modification of the light quantity and quality by inducing a reduced light quantity reaching the canopy, scattering it, and spectrally manipulating it, being dependent on the net colour [22]. Our study tested the influence of three different photoselective nets with similar mesh sizes but different nominal shading factors (4% for the yellow, 7% for the pearl, and 19% for the grey). As a result, in line with the findings of [21], the shading effect could be responsible for the decrease in the pollen fertility observed under the nets, due to a reduction in the total radiation. However, the differences observed were significantly pronounced for the net with the highest shading capacity, the grey one, pointing out that the quality of the radiation reaching the canopy and the shading effect could probably be responsible for the pollen fertility alterations observed.

The lower fertility associated with a larger pollen size can also impair the pollination efficiency and impact the production/fruit size levels, even though this fertility decrease under the nets could be compensated for through artificial pollination and pollinators. A study developed in the same orchard for assessing the influence of the PNs on the yield reported a 40% decrease in the fruit production under the three nets in 2020 compared to the control [23]. One interesting aspect to look at in the future is whether the plants could naturally balance the loss of pollen fertility by producing a greater number of pollen grains per anther, as observed in other species [24].

The biochemical analysis of pollen from the different experimental conditions points to a significant overall cultivar-dependent outcome influenced by the different PNs. For the Matua, the TSP and TSS of the pollen from the PNs decreased compared with the open field conditions; in the Tomuri, an increasing trend was observed. However, no statistically significant differences between the cultivars were observed in the total soluble protein (TSP) and sugar (TSS) content. The PNs only presented a significant influence on the TSS of the pollen. These different TSP responses have been observed in other species when exposed to environmental stressors like air pollution [25] and can lead to differential tolerances of each cultivar studied. Indeed, pollen grains synthesize a set of proteins and biochemical components involved in several vital processes, such as protecting the cellular machinery in response to environmental conditions [17,26], and it has been pointed out that PNs influence the canopy microclimate [22].

Regarding the TSS content of the pollen, a significant influence of the pearl net was observed, which induced an increase in the sugar content of the Tomuri but no change in the Matua. The alteration of the microclimate conditions under the nets can potentially cause a reduction in the stress conditions, such as a higher average soil water content and a decrease in the temperature. This could, in turn, lead to an increase in photosynthetic efficiency that will contribute to the synthesis and availability of carbohydrates in the whole plant, including the pollen [22]. As a result, with its ability to reflect and diffuse much light, the pearl net could have a more pronounced effect, with the Tomuri cultivar being more sensitive. The multiple co-occurring factors associated with the protection of the PNs [20] could also influence the pollen’s TSP and TSS content. A further detailed proteomic/metabolomic study could disentangle the potential effects of PNs on the pollen’s biochemical content, which may affect the pollen’s functions, performance, and possible nutritional value for pollinators.

Raman spectroscopy can detect minor chemical differences within the pollen wall related to the growth conditions [27]. In our study, we observed shifts at some bands of the Raman spectra of the pollen from the plants grown under the PNs and, therefore, under different light conditions [20]. These shifts are indicators of differences in the vibration modes of the molecules, but there were no overall changes at the molecular structure or organisational level. These shifts probably result from the light modifications promoted by the PNs, which allow the scattering of less ultraviolet (UV) radiation than photosynthetically active radiation. It was identified in [22] that a pearl net has the highest light-scattering potential, while the grey one has the lowest. In our study, we observed that the pearl net induced the greatest Raman shift change overall, while the grey PNs the lowest. Investigating whether these differences might have physiological implications such as on pollen–stigma interactions could be interesting.

## 4. Materials and Methods

### 4.1. Study Location and Experimental Design

The experiment was conducted in São Salvador de Briteiros, Guimarães, northwest Portugal (41°30′57.1″ N, 8°19′22.8″ W), at a 30-year-old commercial orchard of *A. deliciosa* cultivars. Hayward plants intercalated with male pollinator cultivars (Matua and Tomuri). All the plants were set in rows with a northwest–southeast orientation and trained and pruned in the pergola type, with wires 2 m above the ground. Yellow, pearl, and grey high-density polyethylene (HDPE) PNs (Iridium^®^, Agrintech, Eboli, Italy) with 2.4 × 4.8 mm sized mesh and a nominal shading degree of 4%, 7%, and 19%, respectively, were installed at a height of 4 m (Figure 3).

The orchard’s soil pH was ~5.6, had an organic matter content of 3.5%, P_2_O_5_ of 122 mg kg^−1^, and K_2_O of 297 mg kg^−1^. The average temperature registered during the study period was 18 °C with a maximum average temperature of 24 °C and a minimum average temperature of 12 °C, and the monthly average precipitation was 100 mm (data accessed at www.ipma.pt).

### 4.2. Pollen Sampling

Pollen samples were collected from the pollinators ‘Matua’ and ‘Tomuri’. The plants were recognised using an identification guide [28]. In each experimental condition (uncovered control and yellow, pearl, and grey PNs), the male flowers from three randomly selected plants, avoiding the parcels’ periphery, were collected in 2020 at full flowering from all quadrants at various heights of each canopy and pooled per condition and cultivar (100 flowers in each pool). Following separation from the other floral components, the anthers were dried at 25 °C for 24 h. The pollen naturally released from the anthers was separated by passing through two different grades of sieves (150 µm and 63 µm), and the samples were stored at −20 °C until the analysis [29].

### 4.3. Pollen Morphological Analyses

The morphometric parameters of the pollen grains were analysed with a transmitted light microscopy (Leitz Laborlux K microscope, Ernst Leitz Wetzlar GmbH, Wetzlar, Germany) at ×400. The length (P), width (E), P/E ratio, and area (A) of 100 pollen grains per experimental condition and cultivar were measured using the ImageJ version 1.46r software (National Institute of Health, Bethesda, MD, USA). The pollen grains were directly dispersed on the microscopic slide and observed.

The pollen exine’s microperfuration density was determined with a scanning electron microscopy (SEM). Each pollen sample, without any pre-processing treatment, was spread on adhesive carbon tape mounted on a pin and coated with an Au/Pd thin film using the SPI Module Sputter Coater equipment (SPI Supplies, West Chester, PA, USA) by sputtering for 120 s and with a 15 mA current. The images were obtained at a magnification of ×20,000 using a JEOL JSM 6301F high-resolution instrument (JEOL, Tokyo, Japan) operating at an accelerating voltage of 15 kV and a working distance of 15 mm. The exine’s mean microperfuration density of each pollen was measured in two 4 µm^2^ fields. Ten pollen grains per condition and cultivar were measured.

### 4.4. Pollen Viability and Germination Assays

The fluorescein diacetate (FDA) staining methodology [30] was used to assess the pollen viability, using a modification of the protocol of Luria et al. [31]. The pollen (5 mg) was suspended in 1 mL of PBS containing FDA (1 µL/mL). After 10 min of incubation in obscurity, the pollen grains were counted in a BD Accuri™ C6 Flow Cytometer (BD Biosciences, Fremont, CA, USA). In the experiment, 488 nm laser excitations and the FITC emission detector (533/30 nm) were used. The core diameter and flow rate were configurated as 40 µm and 100 µL/min, respectively, and 50,000 events were detected per run, which allows assessing the pollen viability in a high throughput automated analysis. The size and granularity of the events were given by FSC-A and SSC-A, respectively, allowing empirical discrimination and gating of the pollen events. An average of approximately 35,000 pollen grains was counted in each run.

Unstained samples (pollen suspended in PBS without FDA) were used for the autofluorescence threshold. Three technical replicates, with three measurements each, were performed per experimental condition and cultivar.

For the germination assays, the pollen (1 mg mL^−1^) was suspended in a culture medium (of 10% sucrose, 0.4 mM of boric acid, and 1 mM of calcium nitrate) and incubated at 27 °C for 2 h to 3 h in a thermo-controlled dryer with soft shaking, according to Abreu and Oliveira [32]. The germination percentage was scored by counting at least 300 pollen grains in three regular-spaced traverse rows under a light microscope (Leica DMLS, Leica Microsystems GmbH, Wetzlar, Germany) at ×400. Only the pollen grains with tubes longer than the grain’s diameter were considered to be germinated.

### 4.5. Total Soluble Protein and Sugar Contents

For the extraction of the soluble proteins, 25 mg of pollen was disrupted, using a Mini-Beadbeater™ (Biospec, Bartlesville, OK, USA) in microtubes with phosphate-buffered saline (1:20 *w*/*v*) with a pH of 7.4, 30 mg of zirconia beads (0.5 mm of diameter), and the cOmplete™ Mini EDTA-free Protease Inhibitor Cocktail (Roche, Mannheim, Germany) at 4 °C, according to Pereira et al. [25]. The lysed samples were kept under constant orbital stirring for 2 h and centrifuged twice at 7000× *g* rpm for 20 min (4 °C). After the supernatant filtration using a 0.45 µm Millipore filter, the filtrate was centrifuged again following the conditions described beforehand. The colorimetric Bradford [33] method was employed to quantify the soluble proteins of the pollen using the Coomassie Plus™ Protein Assay Reagent (Thermo Scientific, Waltham, MA, USA) in 15× diluted extracts. The absorbance was read using a microplate spectrophotometer (Thermo Scientific Multiskan GO, Ratastie, Finland) at 595 nm. A bovine serum albumin (BSA) standard calibration curve was used to estimate the soluble protein concentration of the pollen. Three extraction replicates, with three absorbance measurements for each, were performed per experimental condition and cultivar.

For the determination of the total soluble sugar (TSS), the anthrone–sulphuric acid method described by Irigoyen et al. [34] was adapted for the pollen analysis. The pollen (25 mg) was disrupted, using a Mini-Beadbeater™, in microtubes with ethanol 80% (*v*/*v*) and 30 mg of zirconia beads (0.5 mm), employing the shaking procedure described for the soluble protein extraction. The extract volume was adjusted to 10 mL with ethanol 80% and heated at 80 °C for 1 h. The colorimetric reaction was performed by incubating the supernatant with a freshly prepared anthrone reagent (1:25 *v*/*v*; 80 mg of anthrone: 2 mL of H_2_O: 40 mg of H_2_SO_4_ 96%) at 100 °C for 10 min. After cooling in ice, the absorbance measurements were performed using a microplate spectrophotometer (Thermo Scientific Multiskan GO, Ratastie, Finland) at 625 nm. For the estimations of the TSS, a D-glucose standard calibration curve was used. Three extraction replicates, with three absorbance measurements for each, were performed per experimental condition and cultivar.

### 4.6. Raman Spectroscopy

The Raman spectroscopy analysis was performed following the procedure described in Pereira et al. [25], using an XploRA™ Raman microscope (Horiba Scientific, Palaiseau, France).

The pollen samples were kept at room temperature for 10 min before the analysis, and the spectra acquisition was made with an ×100 objective, at an excitation wavelength of 785 nm, from a diode laser at a power of 25 mW and a range of diffraction gratings with 1200 mm^−1^ lines and a slit of 300 μm. The Raman spectrum wavenumber was calibrated before each measurement using a Si referent standard (520.6 ± 0.1 cm^−1^).

Per experimental condition and cultivar, three spectra per pollen grain (five scans of fifty seconds each) were acquired in 10 different pollen in a spectral region of 800 to 1950 cm^−1^ with a resolution of approximately 1 cm^−1^. The Labspec 6 integrated with KnowItAll^®^ 2020 software was used for the spectral acquisition, and there was pre-processing using an automatic polynomial baseline correction followed by normalisation to a constant area under the curve of 100 (a.u.). The bands’ wavenumber (W) was determined by deconvolution using a mixed Gaussian–Lorentzian curve fitting procedure for 25 bands, which corresponds to the aggregate of the principal bands present in the distinct pollen spectra.

### 4.7. Statistical Analysis

For each cultivar, the means between the experimental conditions were compared by a two-way ANOVA. The Duncan post hoc test was applied whenever significant differences were found. The data normality and homoscedasticity were verified using the Kolmogorov–Smirnov test or the Shapiro–Wilk test (when *n* < 30).

For the spectral analysis, the average spectra for each sample were calculated and compared. A one-way ANOVA followed by a Duncan post hoc test was applied per cultivar to test the hypothesis of significant differences in the prominent Raman peak’s (W) mean values between the experimental conditions. A significance level of 0.05 was assumed except in the Raman data analysis, where 0.01 was used due to the spectra complexity. All the analyses were performed using Statistica™ v14 (TIBCO Software Inc., Palo Alto, CA, USA).

## 5. Conclusions

Our study showed that the PNs could influence several pollen traits, increasing its size, decreasing its fertility, and altering its protein and sugar content. Overall, the pearl PN was the one inducing the greater modifications. It was also observed that the cultivar might influence the outcome of some traits. How these modifications might correlate with the pollinator’s response, the crop production, and the fruit quality, as well as the possible development of compensatory or regulatory mechanisms by the plants, will be interesting to address.

## Figures and Tables

**Figure 1 plants-13-01691-f001:**
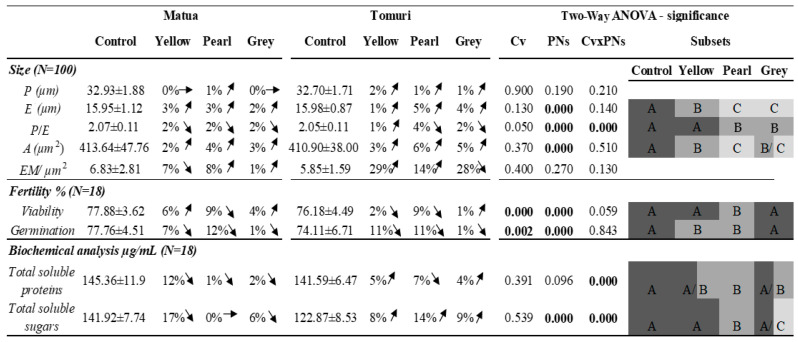
Morphological and ultrastructural parameters, pollen viability and germination, total soluble proteins (TSPs) and total soluble sugars (TSSs) from Matua and Tomuri cultivars grown in open field conditions (the control) and under PNs (yellow, pearl, and grey). Different subset columns colour and letters indicate statistically significantly different means according to the two-way ANOVA test followed by the Duncan post hoc test (*p* < 0.05) for pairwise comparisons of the P, E, P/E ratio, area (A), and exine microperforations (EM). Bold numbers report statistically significant differences. Cv: cultivar; PNs: photoselective nets.

**Figure 2 plants-13-01691-f002:**
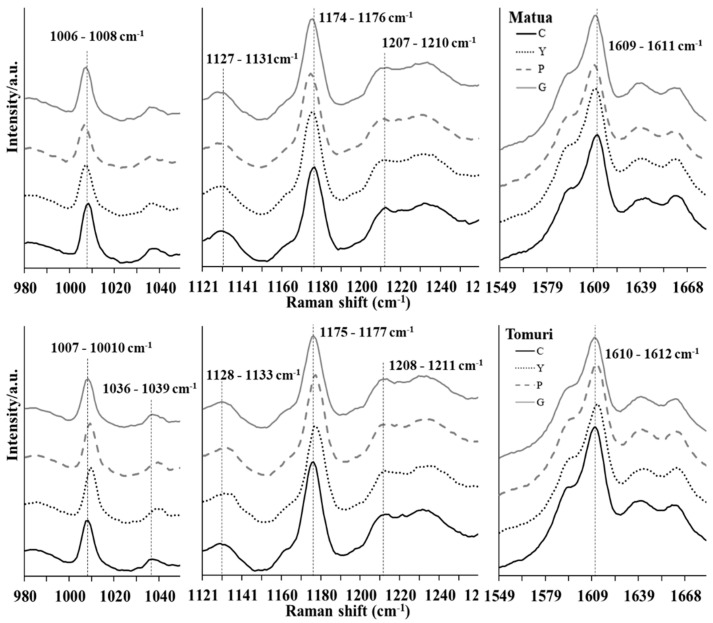
Detailed Raman spectra of Matua (upper) and Tomuri pollen collected in open field conditions (C: control) and PNs (Y: yellow; P: pearl; and G: grey). Prominent band modifications are highlighted (the pointed line).

**Figure 3 plants-13-01691-f003:**
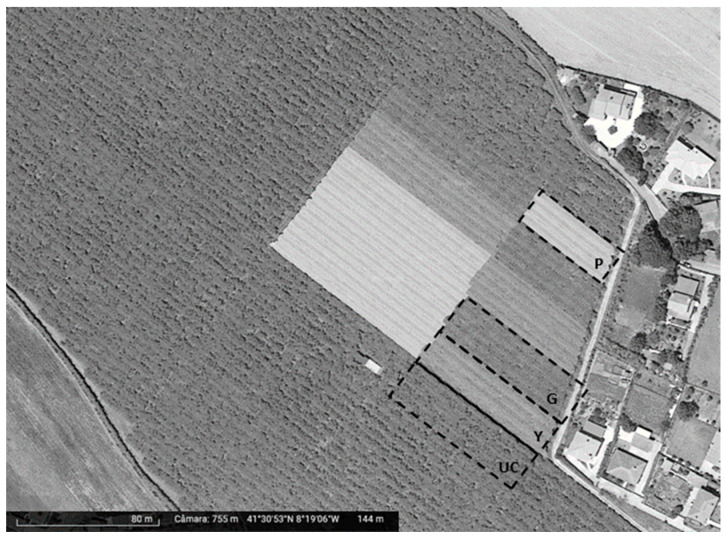
Google, Digital Globe images from the study orchard of *A. deliciosa* Hayward cultivars in São Salvador de Briteiros, Guimarães. Identification of the photoselective nets (PNs): yellow (Y), pearl (P), grey (G), and uncovered control (UC).

## Data Availability

The data are contained within the article.

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
