# Peer review of "Can Photoselective Nets’ Influence Pollen Traits? A Case Study in ‘Matua’ and ‘Tomuri’ Kiwifruit Cultivars"

_plants, 2024, doi:10.3390/plants13121691_

Round 1
Reviewer 1 Report
Comments and Suggestions for Authors
the paper is well written. I have no comments or suggestions
Author Response
We would like to thank the reviewer for the time taken to review our manuscript.
Reviewer 2 Report
Comments and Suggestions for Authors
Dear author, I read the manuscript entitled:
'Can photoselective nets' colour influence pollen traits? A case 2 study in 'Matua' and 'Tomuri’ kiwifruit cultivars'
The text is well written. Of course there are some minor improvements that can increase further the quality of the manuscript such as the reconstruction of table 1, but the major flaw of the experiment is the different shading degree of the nets. This means that you have 3 different factors:
Cv with two levels (Matua and Tomuri)
PN with four levels (Control, Yellow, Pearl, Grey) and
Shading with four levels [0%(control), 4%(yellow), 7%(pearl) and 24% (grey)]
You have not taken into consideration the effect of different net shading.
Thus, the differences you may have observed may be attributed to the combination of shading degree and color or the shading degree of the net per se rather than solely the color of the net.
Author Response
We would like to thank the reviewer for taking the time to review our manuscript.
Point 1: there are some minor improvements that can increase further the quality of the manuscript such as the reconstruction of table 1…
Response: Regarding the reconstruction of table 1, the reviewer did not point out what needs reconstruction, so we do not know how to address this comment.
Point 2: the major flaw of the experiment is the different shading degree of the nets. This means that you have 3 different factors: Cv with two levels (Matua and Tomuri); PN with four levels (Control, Yellow, Pearl, Grey) and Shading with four levels [0%(control), 4%(yellow), 7%(pearl) and 24% (grey)]. You have not taken into consideration the effect of different net shading. Thus, the differences you may have observed may be attributed to the combination of shading degree and color or the shading degree of the net per se rather than solely the color of the net.
Response: Concerning the comment on the three different factors, the factor net colour and factor shading, as suggested by the reviewer, comprise the same observations and, therefore, are not distinct groups and is not correct to be considered together in the statistical analysis. But the reviewer has a point when stating that the differences could be attributed to one or several abiotic variables that are influenced by each net compositional characteristics (e.g. these light-regulated physiological responses, microclimate alterations). Many of these variables are colinear and in the orchards, the global influence of the PNs' characteristics prevails. We, therefore, considered the effect of colour as a hole rather than each specific variable. Thus, if the reviewer does not oppose, we want to maintain the colour of the net as a representative variable.
Reviewer 3 Report
Comments and Suggestions for Authors
Dear Authors,
The findings of this study would be significant for those who are utilizing or are planning to use photoselective nets in agriculture. I have some few questions, comments, and recommendations for the improvement of the paper:
1. Is there a specific reason in choosing yellow, pearl, and grey-colored PNs? Why did you not choose or include others (blue, red) which are commonly used as well?
2. The title of the study is "Can photoselective nets' colour influence pollen traits..." But the discussion on how the PN's colour affects pollen traits is lacking, especially among the colour parameters. The discussion section should be improved, and more reference studies must be provided if possible.
3. Can you provide a supplemental figure for the pollen viability and germination of the cultivars under the different PN, as well as the control?
I attached the manuscript with some minor corrections in grammar, punctuation marks, and typographical errors.

English language is fine. Minor revisions in some words should be done to make the manuscript more scientifically sound.
Author Response
The findings of this study would be significant for those who are utilizing or are planning to use photoselective nets in agriculture. I have some few questions, comments, and recommendations for the improvement of the paper:
We would like to thank the reviewer for taking the time to review our manuscript and for the pertinent and constructive comments.
Point 1: Is there a specific reason in choosing yellow, pearl, and grey-colored PNs? Why did you not choose or include others (blue, red) which are commonly used as well?
Response: We made our study in a commercial orchard, under a research project with the owner's company. So we studied the nets already installed, which correspond to the most widely used in other orchards from the region.
Point 2: The title of the study is "Can photoselective nets' colour influence pollen traits..." But the discussion on how the PN's colour affects pollen traits is lacking, especially among the colour parameters. The discussion section should be improved, and more reference studies must be provided if possible.
Response: The authors thank the comment and agree that there is some lack of discussion on the mechanisms altered during pollen formation that lead to the results obtained in this article. It was our most challenging task to write a comprehensive discussion since, to our knowledge, this is the first study to ever investigate pollen from plants grown under photoselective nets. Therefore, there is no scientific bibliography or analysis focusing photoselective nets mode of action on pollen development, physiology or metabolism to support our results. Nonetheless, the following discussion was added based on the possible effects on the crop due to light and microclimate changes done by the nets:
“Our results point out a fertility reduction under the nets, being significant in pearl (viability and germination) and yellow (germination) PNs, with the cultivar having no significant influence on the outcome of the different PNs nets. An association has been reported between reduced male fertility and reduction in the incident radiation in the anthesis preceding months in wheat but no effect from air temperature [21]. The use of PNs allow light quantity and quality modification by inducing reduced light quantity reaching the canopy, scattering it, and spectrally manipulating it, being dependent on the net color [22]. Our study tested the influence of three different photoselective nets with similar mesh sizes but different nominal shading factors (yellow of 4%, pearl with 7% and grey with 19%). So in line with the findings of [21], the shading effect could be responsible for the decrease in pollen fertility observed under the nets, due to a reduction in total radiation. However, the differences observed were significantly pronounced for the net with the highest shading capacity, the grey one, pointing out that the quality of the radiation reaching the canopy and the shading effect could probably be responsible for the pollen fertility alterations observed. “
“Regarding TSS pollen content, a significant influence of the pearl net was observed, which induced a higher pollen sugar content increase in Tomuri or no change in Matua. The alteration of microclimate conditions under the nets can potentiate the reduction in stress conditions, such as higher average soil water content and a decrease in tempera-ture. This could in turn lead to an increase in photosynthetic efficiency that will con-tribute to the synthesis and availability of carbohydrates in the whole plant, including pollen [22]. So, with its ability to reflect and diffuse much light, the pearl net could have a more pronounced effect, with Tomuri cultivar being more sensitive.”
“Raman spectroscopy can detect minor chemical differences within the pollen wall related to growth conditions [25]. In our study, we observed shifts at some bands of the pollen Raman spectra from plants grown under the PNs and, therefore, under different light conditions [20]. These shifts are indicators of differences in the vibration modes of the molecules but no overall changes at the molecular structure or organisation level. They can probably result from the light modifications promoted by the PNs, which allow the scattering of less ultraviolet (UV) radiation than photosynthetically active radiation. It has been reviewed in [22] that the pearl net has the highest light-scattering potential while the grey one has the lowest. In our study we observed that the pearl net induced overall the greatest raman shift change while the grey PNs the lowest.”
Point 3: Can you provide a supplemental figure for the pollen viability and germination of the cultivars under the different PN, as well as the control?
Response: The viability was tested by flow cytometry, and therefore, there was no picture of pollen. Regarding germination, we did not take any photos, it was a failure on our side.
Point 4: I attached the manuscript with some minor corrections in grammar, punctuation marks, and typographical errors.
Response: Thank you very much for the careful reading. All changes were done.
Reviewer 4 Report
Comments and Suggestions for Authors
Comments to authors
Line 25: “Data also support that the cultivar affects the outcome of some characteristics..” Which data would you please clarify?
Please mention the variability in exine sculpturing as well. The authors only mention variations in pollen size.
Line 55-69: The authors should need to modify this paragraph, as the authors mentioned “the influence of covering nets, such as 58 PNs, on pollen aspects such as its morphology, fertility, and molecular composition still 59 needs to be characterized…” but did not provide background study about the effect of environmental factors on the pollen morphology.
Line 70: hypothesise?
Line 79: “m and area between 410 and 435 µm2” whats means by the area between? Please use the correct terminology for easy understanding. Are the authors talking about mesocolic diameter?
(L/W ratio around 2 µm)? or P/e ratio? Please correct this all over the manuscript.
Line 83: Confirm the term area.
Line 94-103: The authors only focused on the effect of the net on the quantitative features of the pollen. Please provide variability in the qualitative features as well.
Line 211: Please provide the database name that the authors follow for identification of the species, such as World Flora Online.
Line 215-216: “the anthers were dried at 25 °C for 24 h.” I am worried why the authors provide temperature. Because changes in pollen volume and shape during the dehydration phase may be due to harmomegathy. So, how the authors claim that changes in pollen size may be due to the nets?
Line 225: “The pollen exine’s microperfuration density was determined by scanning electron microscopy (SEM)” However, I did not see these variations in the abstract or the results section. Please clarify. Does net effect exine sculpturing? i.e perforate is common, any other ornamentation?
Line 225-228: Please reorganize and arrange them systematically. First, mention the sample preparation methods, then sputtering and finally SEM micrographs…..
Line 282-283: The pollen sample preparation method for RS was not clear. Please clarify each point to make it easy for the readers.
Figure S1: All the pollen is dehydrated as the authors provide temperature for 24 hours. the authors should need to analyze the fresh pollen, for a more clear understanding of the micromorphological features. The colpus surface oranmanetion was also not visible here due to harmomegathy.
If the authors have pollen samples, then please provide the optical microscopy images of the polle as well, to identify the variation in exine and intine thickness.
Comments on the Quality of English LanguageMinor editing of English language required
Author Response
We would like to thank the reviewer for taking the time to review our manuscript and for the pertinent and constructive comments, particularly within the palynological analysis.
Point 1: Line 25: “Data also support that the cultivar affects the outcome of some characteristics..” Which data would you please clarify?
Response: The sentence was changed to : ”Our results also…”
Point 2: Please mention the variability in exine sculpturing as well. The authors only mention variations in pollen size.
Response: The authors did not mention this in the abstract because it was not statistically significantly altered, and we thought that was not important information to refer to. Since there is no significant variability and due to the limitation of the number of words in the abstract, we decided not to describe the exine sculpture.
Point 3: Line 55-69: The authors should need to modify this paragraph, as the authors mentioned “the influence of covering nets, such as 58 PNs, on pollen aspects such as its morphology, fertility, and molecular composition still 59 needs to be characterised…” but did not provide background study about the effect of environmental factors on the pollen morphology.
Response: The authors apologise for not agreeing with the reviewer, but in our opinion, we presented two review articles to back up the effects of environmental factors on several traits of pollen, references [15] and [16].
Point 4: Line 70: hypothesise?
Response: Thank you for the correction; the spelling was corrected.
Point 5: Line 79: “m and area between 410 and 435 µm2” whats means by the area between? Please use the correct terminology for easy understanding. Are the authors talking about mesocolic diameter?
Response: Here, we are reporting the total area of the pollen (total particle area) and not the diameter. The authors changed the sentence to: “…16 µm and total area occupied by the pollen between…”
Point 6: (L/W ratio around 2 µm)? or P/e ratio? Please correct this all over the manuscript.
Response: In the authors' opinion, the polarity of pollen (and therefore the polar and equatorial axes) can only be accurately determined when the pollen is in tetrad form during pollen development. Since the authors did not observe this feature, we opted to state L and W, which we were sure to measure.
Point 7: Line 83: Confirm the term area.
Response: the authors changed to “total area occupied by the pollen”
Point 8: Line 94-103: The authors only focused on the effect of the net on the quantitative features of the pollen. Please provide variability in the qualitative features as well.
Response: The authors apologise but did not understand which “qualitative features”, can the reviewers be more specific. Of course, several other pollen traits could be statistically analysed, we just tested some of them. We would also like to point out that we aimed to investigate possible influences from the nets, and therefore, we wanted to quantify every trait we measured for statistical analysis.
Point 9: Line 211: Please provide the database name that the authors follow for identification of the species, such as World Flora Online.
Response: We conducted our study in a commercial orchard as part of a research project with the owner's company. So, there was not the need to confirm the species.
Point 10: Line 215-216: “the anthers were dried at 25 °C for 24 h.” I am worried why the authors provide temperature. Because changes in pollen volume and shape during the dehydration phase may be due to harmomegathy. So, how the authors claim that changes in pollen size may be due to the nets?
Response: The authors claim that because all pollen grains were exposed to the same conditions, the difference was the nets and conditions they created.
Point 11: Line 225: “The pollen exine’s microperfuration density was determined by scanning electron microscopy (SEM)” However, I did not see these variations in the abstract or the results section. Please clarify. Does net effect exine sculpturing? i.e perforate is common, any other ornamentation?
Response: A sentence was added reporting no statistically significant differences in exine perforation
Point 12: Line 225-228: Please reorganise and arrange them systematically. First, mention the sample preparation methods, then sputtering and finally SEM micrographs…..
Response: The authors apologise but did not understand this comment. In the section on SEM analysis, we first describe the sample coating, the sputtering, and the SEM micrographs, in the order mentioned by the reviewer. Possibly, it is our mistake, and we are not interpreting the comment as we should have.
Point 13: Line 282-283: The pollen sample preparation method for RS was not clear. Please clarify each point to make it easy for the readers.
Response: The authors describe the procedure in the following sentence of the text but agree that it is unclear to the readers, leading to the idea that some parts of the methodology are missing. Therefore, we changed the text to: “Raman spectroscopy analysis was done as described in Pereira et al. [23] using an XploRA™ Raman microscope (Horiba Scientific, France) as followed. Pollen samples were kept at room temperature »
Point 14: Figure S1: All the pollen is dehydrated as the authors provide temperature for 24 hours. the authors should need to analyse the fresh pollen, for a more clear understanding of the micromorphological features. The colpus surface oranmanetion was also not visible here due to harmomegathy.
Response: It was not the authors' intention in this study to detail the pollen morphology. If so, as very accurately pointed out by the reviewer, we would need to observe pollen in its natural conditions, either collected directly from the anther or sampled airborne. We aimed to use quantitative traits of pollen that could be statistically compared among the different experimental conditions.
Point 15: If the authors have pollen samples, then please provide the optical microscopy images of the polle as well, to identify the variation in exine and intine thickness.
Response: The authors believe these features could only be accurately determined using TEM, particularly for the intine. We did not perform this analysis.
Reviewer 5 Report
Comments and Suggestions for Authors
I checked your manuscript and described comments below.
Actinidia deliciosa is an important fruit native to China and eaten all over the world.
This paper provides a very good analysis of the total soluble proteins (TSP) and carbohydrates (TSS) of Actinidia deliciosa.
I think you should consider the following points.
1. I don't know what kind of fruit 'Matua' and 'Tomuri' kiwifruit are. I think it would be better to post these photos.
2. Ref.1 is not written correctly. It should be written in detail.
I don't think this paper has new various major mistakes or grammatical problems.
Author Response
Actinidia deliciosa is an important fruit native to China and eaten all over the world. This paper provides a very good analysis of the total soluble proteins (TSP) and carbohydrates (TSS) of Actinidia deliciosa. I don't think this paper has new various major mistakes or grammatical problems.
We would like to thank the reviewer for taking the time to review our manuscript and for the comments.
Point 1: I don't know what kind of fruit 'Matua' and 'Tomuri' kiwifruit are. I think it would be better to post these photos.
Response: 'Matua' and 'Tomuri' are male pollinizers so we do not have photos from fruits.
Point 2: Ref.1 is not written correctly. It should be written in detail.
Response: reference was corrected as followed: FAO. 2021. World Food and Agriculture – Statistical Yearbook 2021. Rome., thank you.
Round 2
Reviewer 2 Report
Comments and Suggestions for Authors
Please omit the word "Colour" from the title.
Author Response
Change was done and the word "Colour" was deleted from the text.
Thank you.
Reviewer 4 Report
Comments and Suggestions for Authors
The authors revised the manuscript well but still did not respond to all the suggested queries
Line 80: prolate shape (L/W ratio around 2 µm).
The P/e ratio means polar to equatorial ratio mostly used in pollen terminology. So, please change L x W ratio to the P/e ratio. Secondly don’t use the term around, provide actual value.
Secondly, I mentioned that if possible please provide the light microscopic (LM) micrographs of the pollen if the authors have pollen samples. So, by doing this, one can easily see variation in exine thickness. I understand the TEM is costly and not easily accessible, so no need for the TEM here. But I suggested the easy one “the light microscope”
The authors replied, “in our opinion, we presented two review articles to back up the effects of environmental factors on several traits of pollen, references [15] and [16]”. This is good but not sufficient because this is the most important part of the introduction to justify the title. I am talking about the effect of environmental factors on the pollen morphology. Did the authors search for more literature about this? For instance, please refer to the following article; https://doi.org/10.3390/f13050651 and search for other literature as well.
Line 250-251: The authors did not provide the pollen micrographs observed under light microscopy. Please provide it. All the SEM micrographs of the pollen seem dehydrated pollen because of the harmomegathy phenomena. It would be better if the authors investigated fresh hydrated pollen so one can easily see plasticity in pollen features.
Line 250: “Pollen grains morphometric parameters were analyzed by transmitted light microscopy” Please provide the procedure the authors follow for making pollen samples for light microscopy. How do the authors prepare samples for this analysis? Acetolysis methods? Use any chemicals for rupturing the anther?
Line 255: “scanning electron microscopy (SEM)”… provide the company name and model number of the SEM.
Line 256 “Samples were coated with an Au/Pd thin film using the SPI Module..” I think the authors missed some initial steps here. How did the authors put the pollen sample on a metallic stub? Used any adhesive tape for the attachment of pollen? Please carefully provide the details of the methodology, the author follows so the readers can easily understand.
Comments on the Quality of English LanguageMinor editing of English language required
Author Response
Response to Reviewer 4 Comments
We would like to thank the reviewer for taking the time to re-review our manuscript.
Point 1: Line 80: prolate shape (L/W ratio around 2 µm).
The P/e ratio means polar to equatorial ratio mostly used in pollen terminology. So, please change L x W ratio to the P/e ratio. Secondly don’t use the term around, provide actual value.
Response: The authors changed to P/E ratio.
Point 2: Secondly, I mentioned that if possible please provide the light microscopic (LM) micrographs of the pollen if the authors have pollen samples. So, by doing this, one can easily see variation in exine thickness. I understand the TEM is costly and not easily accessible, so no need for the TEM here. But I suggested the easy one “the light microscope”
Response: As mentioned in the past review, the exine thickness cannot be accurately and representatively measured for statistical analysis under a light microscope, particularly when observing pollen directly placed on the slide without any treatment. We have done past studies on pollen morphology of several cultivars of Olea europaea, Vitis vinifera and inclusive Actinidia deliciosa and used TEM and SEM for characterizing the wall (exine and intine). LM was only used to measure the size (L and W), the only feature we can accurately measure on random pollen grain viewed in the microscope field.
Point 3: The authors replied, “in our opinion, we presented two review articles to back up the effects of environmental factors on several traits of pollen, references [15] and [16]”. This is good but not sufficient because this is the most important part of the introduction to justify the title. I am talking about the effect of environmental factors on the pollen morphology. Did the authors search for more literature about this? For instance, please refer to the following article; https://doi.org/10.3390/f13050651 and search for other literature as well.
Response: The authors, in the first revised version of the paper, discussed how the different climatic features could influence the pollen traits measured, including new bibliography. Concerning the introduction part, the authors would like to stand by their opinion. We presented review studies to back up the information. Reviews that gathered the results of several experimental conditions and resumed what is substantial evidence in their results. So, there is no need to cite particular papers.
Point 4: Line 250-251: The authors did not provide the pollen micrographs observed under light microscopy. Please provide it. All the SEM micrographs of the pollen seem dehydrated pollen because of the harmomegathy phenomena. It would be better if the authors investigated fresh hydrated pollen so one can easily see plasticity in pollen features.
Response: The authors included pictures of the light microscopy. Also, we wanted to see the differences in the pollen corresponding to the dispersed state, not when it hydrates in the stigma.
Point 5: Line 250: “Pollen grains morphometric parameters were analyzed by transmitted light microscopy” Please provide the procedure the authors follow for making pollen samples for light microscopy. How do the authors prepare samples for this analysis? Acetolysis methods? Use any chemicals for rupturing the anther?
Response: The pollen was directly dispersed on the microscopic slide and observed.
Point 6: Line 255: “scanning electron microscopy (SEM)”… provide the company name and model number of the SEM.
Response: The company name is JEOL, and the model number is JEOL JSM 6301F, which is already mentioned in the text.
Point 7: Line 256 “Samples were coated with an Au/Pd thin film using the SPI Module..” I think the authors missed some initial steps here. How did the authors put the pollen sample on a metallic stub? Used any adhesive tape for the attachment of pollen? Please carefully provide the details of the methodology, the author follows so the readers can easily understand.
Response: The authors add, "Each pollen sample, without any pre-processing treatment, was spread on adhesive carbon tape mounted on a pin and coated…”.